# Customized Power Wheelchair Joysticks Made by Three-Dimensional Printing Technology: A Pilot Study on the Environmental Adaptation Effects for Severe Quadriplegia

**DOI:** 10.3390/ijerph18147464

**Published:** 2021-07-13

**Authors:** Hee Dong Shin, Da Hyun Ahn, Hyun Ah Lee, Yun Kyung Lee, Hee Seung Yang, Min Jo, Seul Bin Na Lee, Gwan Su Park, Yun Sub Hwang, Woo Sob Sim, Sung-Jun Park

**Affiliations:** 1Department of Physical Medicine and Rehabilitation, Veterans Health Service Medical Center, Seoul 05368, Korea; sk3qjs@hanmail.net (H.D.S.); andahyun@hanmail.net (D.H.A.); hyuna5575@gmail.com (H.A.L.); lyk_angel@naver.com (Y.K.L.); 2Veterans Health Service Medical Center, Veterans Medical Research Institute, 53 Jinhwangdo-ro, 61-gil Gangdong-gu, Seoul 05368, Korea; min8606@gmail.com (M.J.); llssbbnn@naver.com (S.B.N.L.); 3Center of Prosthetics and Orthotics, Veterans Health Service Medical Center, Seoul 05368, Korea; cpo_park@bohun.or.kr (G.S.P.); hys0168@bohun.or.kr (Y.S.H.); s2ellove@bohun.or.kr (W.S.S.); 4Department of Mechanical Engineering, Korea National University of Transportation, 50 Daehak-ro, Cungju 380702, Korea; park@ut.ac.kr

**Keywords:** self-help devices, three-dimensional printing, power wheelchair, quadriplegia

## Abstract

Background: Power wheelchair joysticks are often available as standardized ready-made products for patients with severe hand dysfunction. However, standardized joysticks have limitations in accommodating the individualized features of hand dysfunctions. Three-dimensional (3D) printing technology has facilitated active research on the development of joysticks that can overcome such limitations. Methods: Four subjects participated in the study to evaluate driving abilities and satisfaction after using the customized joystick for two weeks. Modified power-mobility indoor driving assessment (PIDA), National Aeronautics and Space Administration task load index (NASA-TLX), and psychosocial impact of assistive devices scale (PIADS; Korean version) were employed for evaluation. Results: In patients 1–3, the modified PIDA scores had the highest values in the pre-test and post-test. In patient 4, the modified PIDA score had a higher value in the post-test (mean value = 4) compared to the pre-test (mean value = 3.33). In all patients, the modified PIDA time was lower in the post-test compared to the pre-test. The NASA-TLX and PIADS values indicate that greater satisfaction was achieved through the usage of customized joysticks in the post-test. Conclusions: All patients can improve their power wheelchair driving abilities and achieve greater satisfaction. Clinical Relevance: Three-dimensional printed customized power wheelchair joysticks can offer enhanced driving abilities and satisfaction to patients with limited hand function owing to severe spinal cord injury.

## 1. Introduction

Currently, a significant number of people employ wheeled mobility for daily ambulation [1]. Maintaining mobility is one of the most important prerequisites for improving the quality of life (QOL) [2]. Disabled patients require wheelchairs for daily activities and ambulation, but some disabled patients are incapable of using a manual wheelchair owing to limited hand function due to cervical spinal cord injury [3]. A clinical survey [4] reported that 18% to 26% of non-ambulatory patients who are unable to use a manual wheelchair usually cannot use a power wheelchair.

Furthermore, tending to wheelchair-bound patients with neurological disabilities is often difficult for caregivers. Consequently, providing assistive technologies has been identified as a potential solution to reduce the need for human assistance [5]. However, some challenges include caregiver injuries, caregivers’ anxiety about patient injuries, and accessibility issues that limit where they can visit [6].

Automatic driving can help overcome this challenge; however, excessive reliance on automatic driving may hamper their residual physical functions and cause more serious illnesses in the future [7]. To overcome this disability, patients with limited hand function employ power wheelchairs for daily activities and ambulation. However, quadriplegic patients suffer from severe hand dysfunction; therefore, a power wheelchair cannot adequately aid them. Although power wheelchair joysticks are often available as standardized ready-made products for patients with severe hand dysfunction, there are certain limitations in accommodating the individualized features of hand dysfunctions. Previous studies have devised a method to drive power wheelchairs through a shared control system; however, these systems only intervene in a few pre-defined situations. [8] In addition, driving a power wheelchair is a complicated and long-term process, where operation can differ even under the same conditions [7].

Accordingly, studies to develop personalized joysticks have also been conducted. Dicianno et al. [9] demonstrated the customization of a joystick to an individual user. However, when usage was evaluated, it was revealed that the subjects were not using their own wheelchair and did not address reverse driving, which required a different steering strategy. Further, a study by Riley and Rosen on customized joysticks was limited to patients with tremor disability [10].

Advances in science and technology such as three-dimensional (3D) scanning and printing technologies have facilitated research that can overcome such limitations. Park et al. [11] employed a 3D printed finger splint for the treatment of post-hand burn patients. All patients were satisfied with the 3D printed customized splint, as compared with the poor compliance of the conventional ready-made finger splint. Gabriele Baronio et al. [12] suggested that high-accuracy hand orthosis (including fingers) can be achieved through 3D scanning and printing technology; it has been reported that the personalization of patient treatment in the field of orthopedics and rehabilitation is significantly affected by the diffusion of 3D printers, in particular. Recently, 3D printing technology has been used to image cardiovascular intervention in the field of cardiology. Furthermore, it is widely used for dental implants and prosthesis for amputees [13,14,15], but has yet to be employed for wheelchair joysticks.

This study aimed to develop customized wheelchair joysticks using 3D printing technology that aids quadriplegic patients with severe hand dysfunction, thereby improving their driving performance with power wheelchairs.

## 2. Materials and Methods

### 2.1. Participants

The announcement for subject recruitment was made in outpatient and rehabilitation centers, and rehabilitation treatment rooms. Quadriplegic patients using power wheelchairs were not satisfied with the original joysticks employed in our study. The inclusion criteria were patients with quadriplegia due to spinal cord injury who could drive a power wheelchair with a Mini-Mental State Examination (K-MMSE), Korean version [16] score of 24 or more, those who own a power wheelchair, and those who have difficulty in controlling the device smoothly owing to compromised hand function. The exclusion criteria included those who were able to walk more than 100 m, had cognitive impairment with less than 24 K-MMSE points, had difficulty communicating because of aphasia or speech impairment, or were judged by the medical staff to be in a systemic state that made it impossible to conduct a study. A total of five subjects were included in the recruitment period. Patient discomfort and issues regarding previously used wheelchair joysticks were individually identified. One of the five patients was excluded due to the patient’s refusal because the patient’s general condition was aggravated. Using 3D printing technology, customized power wheelchair joysticks were designed and developed. After using the customized joysticks for two weeks, the patients’ driving abilities and satisfaction with the power wheelchairs were evaluated. Due to the nature of the veterans hospital, most of the inpatients were male, and the recruitment of participants was not based on gender, but only males were recruited.

All study-related procedures were performed in accordance with the ethical standards of the institutional and/or national research committees, and the 1964 Declaration of Helsinki. Ethical approval for the study was obtained from the Veterans Health Service Medical Center, Institutional Review Board (No. 2018-08-005-002). Written informed consent was obtained from all subjects prior to conducting the study.

### 2.2. Research Protocol

A 3D scan was performed using a Drake 3D scanner (THOR3D, Moscow, Russia). A 3D printable model was created using the Mimics innovation suite 3-matic ver.14 (Materialize, Leuven, Belgium) software. HP JET Fusion 4200 (Hewlett Packard (HP), Palo Alto, CA, USA), which uses the multi-jet fusion (MJF) method, was used to create a joystick with PA12 (Hewlett Packard (HP), Palo Alto, CA, USA) material.

For patients 1 and 3, 85H11=5-Pastasil, component A (Ottobock, Duderstadt, Germany) and 85H11=5-Pastasil, component B (Ottobock, Duderstadt, Germany) were used for the first embedding material.

A mold of patient 1’s chin was produced using Pastasil, and the mold was scanned and modeled according to the joystick joint using the 3D printer. In addition to the abovementioned common procedure, a silicone pad was added to the customized product to reduce skin irritation, which can occur when the joystick is operated with the chin. Finally, using 3D printing technology, the part composed of Pastasil and the part made of the silicone pad were combined to create a customized joystick (Figure 1(c1)).

For patient 3, a joystick using a golf ball as a knob was used (Figure 1(a3)), and a shape that could support the entire palm was desired. Therefore, their hand was placed on the joystick in use and molded to support the entire palm. Subsequently, through 3D scanning, modeling, and printing, a customized joystick was created (Figure 1(c3)).

For patients 2 and 4, 3D modeling was performed by measuring the size of the conventional joystick used without 3D scanning.

Patient 2 had previously undergone an operation (Figure 1(a2)). The joystick was positioned between the third and fourth fingers with the back of the hand down. A joystick sized to correspond to finger spacing and height was produced (Figure 1(c2)). In addition, for a previously used joystick, the back of the hand was placed on the top of the joystick when reversing, so the upper part of the customized joystick was made into a concave plate shape so that the back of the hand could be placed sufficiently. Thereafter, through 3D modeling and 3D printing, a customized joystick was created.

In patient 4, with the hand in a neutral state, the pillar of the joystick was placed between the thumb and the index finger, and a square roof was fabricated (T-shaped) to prevent the hand from moving upward. Subsequently, through 3D modeling and 3D printing, a customized joystick was created (Figure 1(c4)).

### 2.3. Outcome Measures

The modified power-mobility indoor driving assessment (PIDA) [17], National Aeronautics and Space Administration task load index (NASA-TLX) [18], and psychosocial impact of assistive devices scale (PIADS) [19] were employed for evaluation. Modified PIDA was evaluated before and after using the customized joysticks, and NASA-TLX, and PIADS were evaluated after using the customized joysticks.

The PIDA is a valid and reliable method designed to assess the indoor mobility of persons who use power chairs or scooters, and those who live in institutions [17]. It has been designed to describe an individual’s mobility status at a single point in time, indicating where and how interventions may be implemented, and to evaluate changeover time. Here, we modified the PIDA, and for reasons of patient function and safety, questions related to doors, toilets, and parking were excluded and simplified (Table 1).

Each item was scored using the PIDA and the length of time measured. A score of 1 indicates that the task was not complete. For example, verbal and/or visual cues or physical assistance may be required. A score of 2 implies bumps, objects, or people in a way that causes or could cause harm to the client, other persons, or objects. A score of 3 indicates that completing a task requires several attempts, speed restriction, and/or bumps walls, objects, lightly (without causing harm). A score of 4 means that the subject was completely independent, i.e., the subject exhibited optimal performances and was able to smoothly and safely perform the tasks in a single attempt. It was assumed that the use of the customized joysticks would yield higher scores and shorter times for completing the tasks.

The NASA-TLX is a multi-dimensional rating procedure that provides an overall workload score based on a weighted average of ratings on six subscales: mental demands, physical demands, temporal demands, performance, effort, and frustration [18,20].

The PIADS is a 26-item self-report questionnaire designed to assess the effects of an assistive device on functional independence, well-being, and QOL. In our study, we adapted the PIADS to the Korean version for patients’ understanding of the questionnaire [21].

The patients’ driving abilities and satisfaction with the power wheelchairs were evaluated before and after using the customized power wheelchair joysticks for two weeks.

Data analyses were performed using SPSS version 18.0 for Windows (SPSS Inc., Chicago, IL, USA). As the study data did not have a normal distribution, the Wilcoxon signed-rank test was used to compare the pre-and post-modified PIDA results.

## 3. Results

Baseline characteristics (Table 2) and the results for each patient (Table 3, Table 4, Table 5; Figure 1) are summarized. Table 3 summarizes the PIDA evaluation results before and after using a customized joystick. For patients 1–3, modified PIDA scores have the highest values (patients can follow instructions easily, accompanied by a safe and successful first trial) in the pre-test and post-test. Furthermore, modified PIDA time measurements indicate time reduction in the post-test as compared to the pre-test. In patient 4, the modified PIDA score was higher in the post-test (mean value = 4) as compared to the pre-test (mean value = 3.33). In addition, the modified PIDA time measurement of patient 4 indicates a greater time reduction in the post-test than the pre-test, compared to the other patients (patient 1–3). This suggested that the customized joysticks could compensate for the different features of the patient’s hand dysfunction and improve their driving performance with power wheelchairs.

The NASA-TLX (Table 4) and PIADS (Table 5) evaluations after using a customized joystick indicate greater satisfaction in the post-test. Statistically, there was no significant difference in the relationship between the variables, items of the modified PIDA test, scores, and time (Table 3).

However, in the case of the statistical result with time, the p-value was 0.067 in the part excluding the 180°-turn, and because there were extremely few samples, this result was thought to have been derived. If the sample size was large, statistically, it may have showed a meaningful result.

The PIADS examined the impact of using an assistive device on subjective well-being. Table 5 summarizes the scores of each of the three subscales of the PIADS. The use of the device did not negatively affect well-being; that is, no scores were below zero.

Patient 1 used their chin to control the wheelchair ready-made joystick, which was spherical in shape (Figure 1(a1)). Therefore, their chin constantly slipped from the surface, causing severe discomfort when driving the wheelchair. The modified PIDA score was the same with the customized joystick (Figure 1(b1,c1)), and the time required to accomplish the tasks was shorter when a customized joystick was used. Among the tasks evaluated, entering the elevator door showed the greatest decrease in time. The results of the NASA-TLX assessment indicate reduced workload and improved performance. Furthermore, the PIADS indicated an increase in self-efficacy and decrease in negative emotional reactions to disability.

Patient 2 operated the power wheelchair by placing a joystick between their third and fourth fingers with their palm facing upward. Based on this, a neatly shaped customized joystick (Figure 1(b2,c2)), similar to the self-modified joystick, was modeled and printed using 3D scanning and printing technology. The modified PIDA score was the same with the customized joystick, and the time required to accomplish the tasks was shorter with the customized joystick. Among the tasks evaluated by PIDA, the task of going down a ramp showed the greatest decrease in time. The NASA-TLX assessment indicated a reduced workload and improved performance. Furthermore, the PIADS assessment indicated increased self-efficacy and decreased negative emotional reactions to disability.

Patient 3 previously modified their wheelchair joystick themselves using a golf ball (Figure 1(a3)). Patient 3 placed their hand on the golf ball and operated a power wheelchair with some conserved muscle strength in the upper arm and forearm. There was no support for their wrist, which caused them discomfort during long-term driving. Consequently, a customized joystick that supports the entire palm (Figure 1(b3,c3)) was modeled and printed using 3D scanning and printing technology. The modified PIDA score was the same with the customized joystick, and the time required to complete the tasks was shorter with a customized joystick. Among the items evaluated, maneuverability showed the greatest decrease over time. The NASA-TLX assessment results indicate reduced workload and improved performance. The PIADS assessment results indicate increased self-efficacy and decreased negative emotional reactions to disability.

Patient 4 used a manual wheelchair with the help of a caregiver and was in the process of changing to a power wheelchair at the time of our study. Notably, patient 4 was unable to hold the conventional joystick (Figure 1(a4)). To overcome this limitation, we created a customized t-shaped joystick, which they held between their thumb and index finger (Figure 1(b4,c4)). The modified PIDA score was improved by using a customized joystick, and the time was shorter than that when a customized joystick was used (excluding 180°-turn). Among the items evaluated, maneuverability showed the greatest decrease over time. The NASA-TLX assessment results indicate reduced workload and improved performance. The PIADS assessment results indicate improved self-efficacy and decreased negative emotional reactions to disability. The modified PIDA score had a higher value in the post-test (mean value = 4) than in the pre-test (mean value = 3.33). The modified PIDA time measurement indicates a time reduction in the post-test as compared to the pre-test. A change in score was only observed in patient 4. It can be speculated that this will be more useful when applying the customized joystick to people who use it for the first time than those who have previously used a power wheelchair.

The PIDA, NASA-TLX, and PIADS values indicate greater satisfaction with the usage of customized joysticks in the post-test (Table 3, Table 4, Table 5).

## 4. Discussion 

Recent developments in measurement technology and electronic motors allow the use of wheel-hub mounted power support motor technology in hand rim wheelchairs, as well as hand cycles [22,23]. This type of power support will be significantly useful for people with temporary injuries to the upper extremities [1]. In addition, patients who are long-term wheelchair users owing to the sequelae of the central nervous system (brain or spinal cord injury) or a peripheral nerve injury can also employ power wheelchairs [24]. These patients cannot push the manual wheelchair owing to their severe upper extremity disability; thus, a power wheelchair can be a suitable alternative. However, power wheelchair joysticks are often available as standardized ready-made products. Consequently, there are certain limitations in accommodating the individualized features of hand dysfunctions accordingly, many studies have been conducted on intelligent wheelchairs using hands-free control systems [25,26,27,28,29]. However, basically all of the predetermined commands were performed; only movement in four directions was possible. Therefore, we focused on utilizing a joystick that was customized according to each patient’s preference and functionality using 3D scanning and printing technology. There was also a study that allowed for more independence by developing a modular alternative wheelchair control system using a 3D printer. However, the presence of a wire required an attendant for the correction and reset buttons, which caused an inconvenience to the patient. In addition, Oliver et al. mounted the joystick control unit on a standard joystick, but our study customized the joystick itself to enable individual application [30]. In our study, patient discomfort and issues regarding their wheelchair joysticks were individually identified. In the manufacturing sector, which is moving towards systems for low-volume production, 3D printing technology is actively used in custom production to reflect individual needs [31]. Through 3D scanning and modeling, customized power wheelchair joysticks were designed and developed.

In our study, the modified PIDA was used to evaluate the subjects’ driving abilities with the customized joystick. Although there was no statistical significance, a time reduction was confirmed in executing each item, and in the case of patient 4, the score was also improved. This suggests that customized joystick using 3D printing technology can improve user driving abilities.

Additionally, we evaluated user workload and performance with the customized joystick using NASA-TLX. In all patients, demand, effort, and frustration levels decreased, and performance improved.

Using PIADS, we also evaluated how the customized joystick affected user competence, adaptability, and self-esteem. All patients scored at least 1 in the subscore, which means positive results in functional independence, well-being, and QOL.

Our results from assessing these parameters show that, after using the customized joysticks for two weeks, the patients improved their power wheelchair driving abilities and experienced greater satisfaction.

Additionally, 3D printing technology is driving major innovations in, among others, engineering and medicine. In medicine, 3D bioprinting is already being used for the creation and implantation of multiple tissues, including multilayered skin, bone, vascular grafts, organ splints, heart tissue, and cartilage structures [31].

Moreover, from an engineering point of view, studies on intelligent power wheelchairs (iChairs) and innovative wheelchair service provision models using 3D printing technology have also been reported [32,33]. In addition, customized seat research using 3D printing technology is also possible. Thus, the applicability of 3D printing technology can be widened to include designing seats and joysticks, providing convenience, stability, and satisfaction to patients with severe disabilities [34].

## 5. Conclusions

The limitation of our study is that the sample size was too small. Further, both pre- and post-tests were performed for the modified PIDA, but only the post-test was performed for the PIADS and NASA-TLX assessments. As such, the evidence for the validity of the conclusions may be limited. In addition, because of the pathological nature of spinal cord injury patients, caregivers are often required; hence, the satisfaction of the caregivers should also be evaluated in the future.

Even with these limitations, research using 3D printing technology has many advantages. First, it is sensitive to individual characteristics and requirements, uses materials that are easy to change, and can be recreated using initial measurement data [35]. This improves the convenience for people with disabilities because they can quickly respond to device failure or damage. In addition, feedback can be immediately received, increasing adaptability to the device.

Additionally, the lightweight joystick fabricated in our study has a simple design that can be customized to individual characteristics. No sensors or batteries are required, and it operates silently. To realize these advantages, this is the first study to evaluate the efficacy, satisfaction, and preference of customized aids fabricated using 3D printers.

## Figures and Tables

**Figure 1 ijerph-18-07464-f001:**
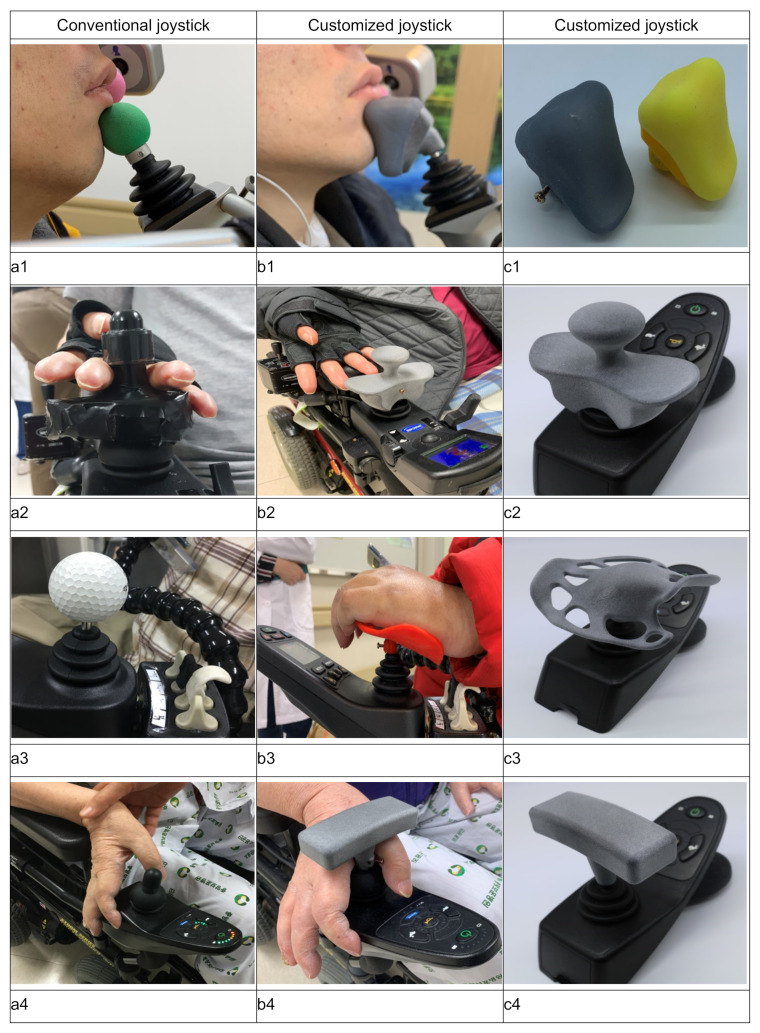
Joysticks employed in this study. (**a**) Conventional joystick. (**b**) Patient using the power wheelchair with a customized joystick. (**c**) Customized joystick.

**Table 1 ijerph-18-07464-t001:** Modified Power-Mobility Indoor Driving Assessment Manual (PIDA).

Modified Power-Mobility Indoor Driving Assessment Manual (PIDA)
1. Accessing Bed from Client’s right Side
2. Entering the Elevator Door
3. Exiting the Elevator
4. Turning right at 4-way intersection
5. Turning left at 4-way intersection
6. 180°-Turn
7. Maneuverability: “Drive in and out between the chairs”
8. Up the Ramp
9. Down the Ramp

**Table 2 ijerph-18-07464-t002:** Patient characteristics.

	Patient 1	Patient 2	Patient 3	Patient 4
Sex	Male	Male	Male	Male
Age (years)	36	73	53	77
Height (cm)	177	167	183	175
ASIA Impairment Scale	A	A	A	ND
NLI	C2	C4	C4	C5 *
Sensory Levels	C2	C4	C4	C5 *
Motor Levels	C2	C5	C4	C5 *
Years Since Injury	16	24	35	12
Etiology	Trauma	Trauma	Trauma	Tumor(cervical) CIDP

* Non-SCI condition. CIDP: Chronic inflammatory demyelinating polyneuropathy. NLI: Neurological level of injury. ND: Not determined.

**Table 3 ijerph-18-07464-t003:** Modified Power Mobility Indoor Driving Assessment (PIDA).

Test		Patient 1		Patient 2		Patient 3		Patient 4	
		Pre	Post	Pre	Post	Pre	Post	Pre	Post
Accessing Bed from Client’s right Side	Score	4	4	4	4	4	4	3	4
	Time (s)	6.35	6.26	10.34	9.45	9.17	7.87	48.38	23.46
Entering the Elevator Door	Score	4	4	4	4	4	4	3	4
	Time (s)	5.24 **	2.72 **	3.56	3.21	4.39	3.13	9.36	2.01
Exiting the Elevator Turning right at 4-way intersection	Score	4	4	4	4	4	4	3	4
	Time (s)	7.87	5.41	7.14	4.34	5.31	4.87	24.85	4.83
Turning right at 4-way intersection	Score	4	4	4	4	4	4	4	4
	Time (s)	3.08	2.92	4.43	3.39	4.78	3.47	10.85	4
Turning left at 4-way intersection	Score	4	4	4	4	4	4	4	4
	Time (s)	3.01	2.55	5.25	3.76	4.49	3.34	6.51	5.38
180°-turn	Score	4	4	4	4	4	4	4	4
	Time (s)	3.67	2.9	7.71	6.78	5.05	4.34	7.68	8.22
Maneuverability	Score	4	4	4	4	4	4	3	4
	Time (s)	7.67	6.82	12.6	10.18	9.65 **	6.61 **	47.81 **	20.45 **
Up the Ramp	Score	4	4	4	4	4	4	3	4
	Time (s)	24.44	24.24	28.41	22.62	24.38	21.85	57.45	39.03
Down the Ramp	Score	4	4	4	4	4	4	3	4
	Time (s)	25.25	23.44	40.31 **	30.77 **	26.63	24.85	63.81	40.97

** The item with the largest difference in time for each patient from among the modified PIDA items.

**Table 4 ijerph-18-07464-t004:** National Aeronautics and Space Administration task load index (NASA-TLX).

NASA-TLX	Patient 1	Patient 2	Patient 3	Patient 4
Mental demand	Low	Low	Low	Low
Physical demand	Low	Low	Low	Low
Temporal demand	Low	Low	Low	Low
Performance	Good	Good	Good	Good
Effort	Low	Low	Low	Low
Frustration level	Low	Low	Low	Low

**Table 5 ijerph-18-07464-t005:** Summary of Psychosocial Impact of Assistive Devices Scale (PIADS).

Participants	Subscale Score		
	Competence	Adaptability	Self-Esteem
Patient 1	3	3	3
Patient 2	1.58	1	1.25
Patient 3	1.33	1	1
Patient 4	1.83	2.33	1.25

The score ranges from −3 to 3, 0 = not any more or less, 1 or 2 = somewhat more, 3 = very much more. Three decimal places rounded off.

## Data Availability

The data used to support the findings of this study are available from the corresponding author upon request.

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
