# Peer review of "Customized Power Wheelchair Joysticks Made by Three-Dimensional Printing Technology: A Pilot Study on the Environmental Adaptation Effects for Severe Quadriplegia"

_ijerph, 2021, doi:10.3390/ijerph18147464_

Round 1
Reviewer 1 Report
The paper discusses the design, manufacturing and user evaluation of reverse-engineered and 3D printed wheel joystick for people with special needs. In general, the literature review and discussion could be more extensive. The limitation of this research - a low number of participants - is acknowledged. However, the discussion could provide further recommendations for research not only from an evaluation perspective but design and manufacturing as well.
There are a few minor typos - mostly around spaces (170, 198) and capital letters (118, 119).
181-184 - The number of participants should be moved to Methods. This number should also be clearly indicated in the abstract.
146 - ㅇㅇ ???
Author Response
We will send you a cover letter file.
Please see the attachment

Reviewer 2 Report
I found this manuscript to be a strong submission with interesting results. This is a well written manuscript that presents the construction and performance of a joystick that was customized according to each patient’s preference and functionality using 3D scanning and printing technology. Through 3D scanning and modeling, customized power wheelchair joysticks were designed and developed. The obvious limitations are highlighted in the paper.
The originality of the paper consists in the customized construction of the wheelchair’s control system. Through 3D scanning and modeling, customized power wheelchair joysticks were designed and developed. Definitely interesting are the identified solutions adapted to each patient’s needs.
The state-of-the-art section is presented clearly and the proposed direction of research are well defined.
The paper layout is generally correct and clear.
The work method is adequately devised and the results are correct. The results would have, however, gained in relevance, had the number of tested patients been greater than 4.
Author Response
We will send you a cover letter file.
Please see the attachment.

Reviewer 3 Report
This work describes the usage of customized 3D printed joysticks to drive a powered wheelchair by 4 individuals with severe disability. The work is pretty simple and straightforward, and the usage of 3D printed handles is definitely something useful for the community of impaired people. However some critical issues must be addressed by the authors before publication.
The limited sample size does not allow any statistical analysis, which results meaningless. It's hard to understand the difference between this work and the state of the art. Lack of baseline assessments of the satisfaction of the participants with their standard joysticks makes the analysis of the results harder to observe.
Last but not least, English is complex in some parts and requires a further revision.
Major Comments:
- It's hard to understand the novelty in this paper, above all when considering that the authors cited another similar work performed by Park et al [11]. Is it just the different clinical population that made a distinction between these two papers?
- Table 4: not clear how you defined "** Items showing the biggest difference". Is that for every activity? And then, why I don't see ** at every line in at lease one cell? Or is it for the whole test? And in this case, how you normalized time difference between activities that lasted ~30s vs other that lasted ~5s?
- Having the NASA-TLX and PIADS only been performed post-test, it's hard to quantify the increased satisfaction to the new joysticks, above all given that for the modified PIDA, 3 out of 4 didn't improve their performance (apart not negligible reduction in time). A baseline execution of these surveys should have been carried out for strengthening the results.
- [line 211] I may agree that the sample size is too small and so results are not statistically significant because of it and not because the analyzed variables are not impacted. But, if the sample size is too small, it's completely useless to run any statistics out of it, because the results will be not enough explicit (like in your case). Remove any statistical analysis or increase the sample size.
Minor Comments:
- Throughout the whole paper, the citations are written after the punctuation (e.g. line 35 "...for daily ambulation. [1]"). Please change them to a more standard style: [citation] punctuation --> "... for daily ambulation [1]."
- [line 82/83] The sentence "Between September 1, 2018, and December, 31, 2018 ... employed in this study" as it is written, leads the reader to believe the those patients were unsatisfied only in the period between September and December. Also this sentence is probably not necessary.
- Section 2.1. should give the reader the number of participants enrolled in the study.
- [line 100/101] "Written informed consent was obtained from all subjects and the study was conducted." --> "Written informed consent was obtained from all subjects before the study was conducted."
- [line 126] "Since patient 2 had previously undergone an operation (Figure. 1-2(a)), the back of the hand was placed between the third and fourth fingers to be operated..." Despite I understand the message of this sentence by looking at the pictures, this sentence has no meaning in English ("back of the hand placed between third and fourth fingers"??).
- Figure 2 is of very poor quality and not necessary/useful to add any information that cannot be already obtained by the picture in figure 2.C.
- [line 142] PIDA is cited twice.
- [line 146] Not sure what are those circles after the [17] citation.
- [line 149/151] "Here, we modified ... parking were excluded" needs English revision. Again, the message is simple and clear enough, the text is badly written.
- Table 2 is not necessary. You are not modifying the original NASA-TLX test, like in the case of the PIDA, so it's not necessary to provide the table. Otherwise, for coherence, you have to give the reader also the PIADS 26 questions.
Author Response

(The authors gave the same response as above.)

Round 2
Reviewer 3 Report
The reviewer thanks the authors for addressing all the requests. Despite the small pool of participants, the paper is now appropriate for publication.